# Navigating persuasive strategies in online health misinformation: An interview study with older adults on misinformation management

**Wei Peng**[1]*, **Jingbo Meng**[2], **Barikisu Issaka**[3]

**1** Department of Media and Information, Michigan State University, East Lansing, Michigan, United States of America, **2** School of Communication, Ohio State University, Columbus, Ohio, United States of America, **3** Department of Advertising and Public Relations, Michigan State University, East Lansing, Michigan, United States of America

\* pengwei@msu.edu

**Data Availability Statement:** The authors are not able to provide any data beyond what is presented in the manuscript due to restrictions that study participants agreed to when they signed the

## Abstract

Online health misinformation commonly includes persuasive strategies that can easily deceive lay people. Yet, it is not well understood how individuals respond to misinformation with persuasive strategies at the moment of exposure. This study aims to address the research gap by exploring how and why older adults fall into the persuasive trap of online health misinformation and how they manage their encounters of online health misinformation. Using a think-aloud protocol, semi-structured interviews were conducted with twenty-nine older adults who were exposed to articles employing twelve groups of common persuasive strategies in online health misinformation. Thematic analysis of the transcripts revealed that some participants fell for the persuasive strategies, yet the same strategies were detected by others as cues to pin down misinformation. Based on the participants' own words, informational and individual factors as well as the interplay of these factors were identified as contributors to susceptibility to misinformation. Participants' strategies to manage misinformation for themselves and others were categorized. Implications of the findings are discussed.

## Introduction

Online health misinformation is health-related information disseminated on the Internet that is false, inaccurate, misleading, biased, or incomplete, which contradicts the consensus of the scientific community based on the best available evidence [1]. Some researchers distinguish misinformation and disinformation by intentionality such that misinformation is unintentionally false information, whereas disinformation is intentionally false information that is spread deliberately to achieve a certain goal [2]. However, many researchers consider misinformation as an umbrella term to describe all false and inaccurate information lacking scientific evidence because it is hard to know whether such information is created and spread intentionally or not [3]. We adopt the second approach to define misinformation in the current study. Online

consent form, which was approved by the Michigan State University IRB.

**Funding:** The study was partially funded by the Brandt Fellowship awarded to Wei Peng. The funders had no role in study design, data collection and analysis, decision to publish, or preparation of the manuscript.

**Competing interests:** The authors have declared that no competing interests exist.

health misinformation carries serious social and public health implications, including links to hesitancy in vaccines, hesitancy in cancer treatment and screening, as well as distrust in science, medicine, and the medical and research communities [4, 5].

Although health misinformation can be traced back to the time of hunting and gathering societies [6], online health misinformation, especially on social media, poses unique challenges due to its use of persuasive strategies, making it more difficult to identify and address [1]. It garners more attention in terms of number of views and shares compared to accurate public health information [7], making it vital to understand how individuals process these strategies, how and why they succumb to them, and how they react to and manage such misinformation. An in-depth understanding of such can provide insights for future misinformation mitigation and debunking efforts.

In the past decade, older adults have significantly grown more reliance on using technology and the Internet for health information according to the US National Health Interview Survey [8]. However, older adults generally have been found to be more vulnerable to online misinformation, with users over 50 responsible for 80% of fake news shares [9]. A review argues three characteristics for older adults' vulnerability to online misinformation, including cognitive decline, social changes, and lack of digital literacy [10]. These characteristics of older adults may make them at risk of stumbling upon persuasive strategies of online misinformation. For example, due to cognitive decline, older adults may rely more on heuristic cues rather than delving into the reasoning when evaluating information. For social change, given that interpersonal trust grows with age [11, 12], older adults may tend to believe vivid stories told by another individual. Therefore, this study will focus on the older adult population.

Existing research on persuasive strategies in online misinformation primarily uses content analysis [13–16], or experiments to examine the effects of these persuasive strategies [17, 18]. In-depth qualitative research, especially during the moment of engagement with misinformation, is limited [19]. Through participants' narratives of their own thought processes, we can gain a better understanding of their reactions as well as how and why they succumb to misinformation. Additionally, information management insights from individuals who recognize these persuasive strategies can inform future digital health literacy programs.

This study aims to bridge this gap by conducting interviews to answer the following research questions (RQ)s: how do older adults respond to misinformation crafted with persuasive strategies? (RQ1), how and why do they fall for such misinformation crafted with these persuasive strategies? (RQ2), and how do they engage in effective misinformation management? (RQ3).

## Literature review

### Persuasive strategies in misinformation

Using language to influence others' beliefs, attitudes, or behaviors is referred to as a persuasion strategy [20]. Health-related misinformation creators employ persuasive techniques akin to those used in prosocial health efforts [21]. A systematic review summarized 12 groups of persuasive strategies in online health misinformation [1].

The first group of persuasive strategies in online health misinformation is fabricating narrative with details–presenting vivid stories that highlight particulars to create online health misinformation [1]. Secondly, misinformation uses anecdotes and personal experiences to spread false information [22]. The strategy of distrusting government sources or pharmaceutical companies constitutes the third group: implying that certain health issues and regulations are being implemented by the government and pharmaceutical companies with the intention of making money [23]. The fourth group is politicizing health issues. This includes the use of

values and ideologies such as freedom, religion, and politicians, and capitalizing on people's propensity to favor members of their own group and be prejudiced against those of other groups [13]. The fifth group highlights the uncertainty and risk associated with medical actions and uses ambiguous and tentative words for audiences to cast doubt on medical actions [15]. The sixth strategy is to undermine science by attacking its innate limitations, such as over-throwing early discoveries with more recent ones [1]. Online health misinformation also inap-propriately uses scientific evidence or uses the evidence out of context [14]. The eighth group is rhetorical tricks that generate a false balance of misinformation, exaggerate some informa-tion, and selectively omit other information [18]. The ninth group employs biased reasoning to draw conclusions. Users of this tactic invoke logical fallacies, misleading linkages, or the combination of unconnected pieces of knowledge, among other strategies, to advance and cir-culate misinformation [24]. The tenth group appeals to people's emotions and not factual information to persuade them [25]. The eleventh group includes linguistic markers such as uppercase words, first-person pronouns among others [15]. Lastly, online health misinforma-tion attempts to establish legitimacy by exploiting a number of well-established tactics, includ-ing citing seemingly credible sources or the use of surface credibility markers to take advantage of users' propensity to process information heuristically [26].

These persuasive strategies were systematically identified from 58 studies. Although prior research examined the effects of some particular persuasive strategies on people's intention to trust or share misinformation [17, 18], these studies were conducted in experiments and lacked in-depth understanding of how people process these persuasive strategies. Thus, RQ1 explores how older adults respond to and process these twelve groups of persuasive strategies embedded in online health misinformation.

## Why people fall to misinformation

The above persuasive strategies are summaries of message characteristics that complicate indi-viduals' ability to identify misinformation, making them more susceptible to it [1]. While sus-ceptibility to misinformation is present in all individuals, some are more prone to accept and share on social media, and fall victim to it [27, 28]. Psychological traits also predict individuals' susceptibility to health misinformation, such as knowledge/skills, cognitive styles, and predis-positions [27]. One widely held belief, sometimes referred to as the "deficit theory," holds that people who accept false information may lack the necessary knowledge or literacy to discern the truth [28]. For instance, prior research [29] discovered that higher education levels improved recognition of online misinformation among older adults. Similarly, individuals who have training in spotting false information are more adept at spotting false health claims online [30].

Compatibility with prior beliefs is another factor that makes individuals susceptible to online misinformation. More specifically, confirmation bias refers to the tendency to believe in the evidence that confirms rather than disconfirms people's existing knowledge, beliefs, and experiences [31]. People tend to accept information aligning with their views. When a new piece of information that is consistent with existing knowledge is accepted, it is very difficult to change, and this resistance to change increases with the size of the congruent knowledge base [32]. Identity-motivated thinking is a unique case of compatibility of belief that supports false information [33]. Identity-threatening information lowers the evaluation of information, which makes people more susceptible to misinformation [34].

The "illusory truth effect" also explains why people believe false information [35]. Fluency, or the ease of processing information [36], often resulting from repeated exposure, may con-tribute to the illusory truth effect. This effect may explain the acceptance of COVID-19

conspiracy despite the lack of any convincing proof, as claims presented multiple times seem more credible [37].

These prior studies offer psychological drivers of health misinformation susceptibility. However, most of these studies are quantitative in nature and lack qualitative research that examines how individuals succumb to misinformation at the moment of encountering it [19, 29], especially using their own words to explain why they believe the misinformation and how they assess its credibility. Thus, RQ2 explores why and how older adults fall for online health misinformation crafted with persuasive strategies.

### Misinformation management in the age of social media

In the age of rampant online misinformation, recognizing and managing false information is challenging for people of all ages [38]. Typically, individuals engage a two-step authentication process: internal and external [39]. People rely on the self (their own wisdom, instinct, and insight), the source, and the message (intrinsic tone and characteristics of the information) as their three primary sources of authentication during the first interaction. After the first stage, if the person is still not persuaded that the information is accurate, he or she moves on to the second stage, which entails performing external acts of authentication such as deliberately or incidentally utilizing interpersonal (friends and families) and institutional resources.

Responses to misinformation vary. Some proactively counteract by commenting to let others know it is false, messaging the poster to caution them, or reporting false posts so it is taken down, while others hesitate due to societal concerns [38]. Fear of being attacked and the desire to maintain social relationships hinder the proactive management of misinformation [40]. For example, people have to consider face-saving and choose appropriate communication channels to discuss misinformation so that others do not feel their self-images are threatened [41]. In addition, they also weigh factors like issue importance, relationship closeness [42], the person's personality [43], and the impact the misinformation has on others to decide whether and how to manage the situation [40]. Specifically for older adults, while some research found older adults are less likely to correct COVID-19 misinformation from others [44], recent research found that older adults are more likely to challenge misinformation in general compared to their younger counterparts on social media [40]. In the current study, RQ3 explores the best practice for managing online health misinformation among older adults as our third research question.

## Method

### Participants

The results reported here were part of a Zoom-based semi-structured interview study conducted during the COVID-19 pandemic in 2021. Recruitment started on August 20, 2021 and ended on October 20, 2021. Only results relevant to the research questions were reported in this paper and other findings were reported elsewhere [45]. Participants were recruited via a local community subject pool and an online participant-sourcing platform, CloudResearch. A screening questionnaire was distributed to include eligible participants who were at least 50 years old, English-speaking, and able to use Zoom. Twenty-nine older adults participated: 20 (69%) were female, and nine (31%) were male. We intentionally oversampled the participants in the racial minority group and lower education levels to have a diverse sample. The screening questionnaire also measured their general health, how frequently they think about health issues (e.g., diet, vaccine, food safety, memory loss, physical activity, cancer prevention), need for cognition, and self-efficacy of online information search. An invitation to sign up for the interview study was sent to a total of 58 participants who completed the screening

**Table 1. Description of participants.**

| Participant | Sex | Year of Birth | Race | Education | Income |
|---|---|---|---|---|---|
| 1 | Male | 1960 | White | Master's degree | $100,000–$149,999 |
| 2 | Female | 1971 | White | 4 year bachelor's degree | $80,000–$89,999 |
| 3 | Female | 1957 | White | Master's degree | $80,000–$89,999 |
| 4 | Male | 1965 | White | Some college but no degree | $90,000–$99,999 |
| 5 | Male | 1951 | White | 4 year bachelor's degree | $100,000–$149,999 |
| 6 | Male | 1962 | White | High school diploma or equivalent including GED | $80,000–$89,999 |
| 7 | Female | 1955 | White | High school diploma or equivalent including GED | $80,000–$89,999 |
| 8 | Female | 1958 | White | Master's degree | $100,000–$149,999 |
| 9 | Female | 1953 | White | Some college but no degree | $10,000–$19,999 |
| 10 | Female | 1963 | White | Some college but no degree | $60,000–$69,999 |
| 11 | Female | 1960 | White | High school diploma or equivalent including GED | $20,000–$29,999 |
| 12 | Female | 1961 | White | Some college but no degree | |
| 13 | Female | 1945 | White | Some college but no degree | Not provided |
| 14 | Female | 1962 | White | Master's degree | $90,000–$99,999 |
| 15 | Female | 1948 | Black/African American | Master's degree | $50,000–$59,999 |
| 16 | Female | 1966 | Black/African American | Some college but no degree | $20,000–$29,999 |
| 17* | Male | 1964 | White | Doctoral degree | More than $150,000 |
| 18 | Female | 1956 | White | Master's degree | $20,000–$29,999 |
| 19 | Male | 1958 | Black/African American | 2 year associate degree | $20,000–$29,999 |
| 20 | Female | 1948 | White | 2 year associate degree | $10,000–$19,999 |
| 21 | Female | 1965 | White | 4 year bachelor's degree | $90,000–$99,999 |
| 22 | Male | 1955 | White | Master's degree | $30,000–$39,999 |
| 23 | Female | 1962 | Black/African American | 4 year bachelor's degree | $100,000–$149,999 |
| 24 | Male | 1960 | White | 4 year bachelor's degree | $90,000–$99,999 |
| 25 | Male | 1962 | White | Professional degree (JD, MD) | More than $150,000 |
| 26 | Female | 1966 | White | Master's degree | $100,000–$149,999 |
| 27 | Female | 1950 | White | 4 year bachelor's degree | $80,000–$89,999 |
| 28 | Female | 1959 | White | 2 year associate degree | $20,000–$29,999 |
| 29 | Female | 1970 | Black/African American | 2 year associate degree | $20,000–$29,999 |

Note.

* Participant identified as Spanish, Hispanic, or Latino.

questionnaire, and 29 participants completed the interview. Table 1 includes the detailed demographic information of the participants.

## Data collection and procedure

The study was approved by the Institutional Review Board of Michigan State University. Before the interview, the participants completed a short online survey to schedule their interview session. A consent form was provided before the online survey started. The participants clicked the "next" button to indicate their consent to participate in this study and to start the online survey. At the beginning of the interview session, the interviewer went over the consent form with the participants and obtained verbal consent once more. No written consent was obtained and recording of the interview started after the verbal consent was obtained. The interviewer documented the verbal consent. Each interview lasted an average of 62 minutes. Participants received $15 in cash or an Amazon gift-card. Participants were first asked about

their perception of health-related misinformation. Then, they were exposed to six pieces of randomly displayed online articles. These articles were written based on online content and incorporated 12 groups of common persuasive strategies identified in online health misinformation. Among the articles, four were misinformation (i.e., cashews can treat depression, sunscreen causes cancer, radiation contaminated pet food, masking not necessary and COVID-19 is just a flu), one was factual information (i.e., the long-term effectiveness of COVID-19 vaccine), and one was uncertain information (i.e., association of caffeine intake and Alzheimer's). For example, in the article about radiation-contaminated pet food, the persuasive strategies of emotional appeals and distinctive linguistics were used in the sentence "BEWARE that the food you purchase for your fur friends can kill them!" S1 Appendix included the annotation of the persuasive strategies in the four misinformation pieces. Uncertain and factual information pieces were mixed into the misinformation pieces to ensure that the participants would not mindlessly consider all information presented as misinformation because they knew this study was about misinformation perception. The results of this study were based on the participants' discussion of the four misinformation articles. S2 Appendix included the six pieces of information presented to the participants. Except for the COVID-19 related messages, participants mainly indicated that they had little prior knowledge of the topics of the message.

The participants were asked to think aloud and vocalize their thoughts as they read [46]. After they read each piece of information and verbalized their thoughts, the interviewer asked them a set of questions based on the interview guide, e.g., "what specific elements of the information made you think that it is believable?", "what actions would you take after reading this article/post?".

## Data analysis

The audio recordings from the interviews were transcribed verbatim. Transcripts were inductively analyzed using the NVivo 12 software. The study employed a 5-stage process of thematic analysis [47], which included becoming familiar with the transcribed interview data, coding, outlining initial themes, reviewing and revising themes, and defining final themes. The first and second authors and two graduate students coded the transcripts. Each transcript was coded independently by two coders. To ensure accuracy, primary and secondary coders analyzed each transcript, resolving any discrepancies through discussion. The nodes were iteratively compared and revised during the coding process. Only nodes and themes relevant to the research questions in this study were reported to focus on the most relevant results while ensuring conciseness. Findings related to information processing and evaluation were reported elsewhere [45]. Finally, the first author reviewed, revised, and finalized themes in NVivo 12.

## Results

### How older adults react to persuasive strategies in online health misinformation (RQ1)

This section illustrates how older adults reacted to the twelve groups of persuasive strategies within these misinformation pieces. Participants' own words were used to explain how these persuasive strategies could both alert some individuals to potential misinformation and trick others into false information.

### Narrative with details

Some individuals were swayed by realistic stories presented in the misinformation, while others used vivid details in it to verify its accuracy. For instance, in reaction to the misinformation

about using the Geiger counter inherited from a professor of physics to check radioactive pet food imported from East Asia after the Fukushima nuclear disaster, P29 found the article credible due to the 'fact' that "they put the meter next to the cat food, and it was three times fold". In contrast, the same detail triggered suspicions for P9, who explained: "the Geiger counter thing, and putting in their can of cat food, I mean there was the reliability of an old device, like that sounds like it's quite old." P8 said: "the inherited Geiger counter is just not, you know, doesn't get it with me. . .. . .well yeah, I think that there are people who would think that "oh my gosh this person measured it with a Geiger counter," you know, "that must be you know it's a machine and it works and it said this.""

## Anecdotes and personal evidence

Anecdotes and personal evidence were considered as red flags for those adept at spotting misinformation. When reading a Facebook post of a woman blaming sunscreen containing paraben for causing breast cancer, P13 responded: "making a really strong statement about the skincare industry based on her personal experience." P14 had a similar reaction: "it's this person's talking about her personal experience." Conversely, P29 fully believed this misinformation, justifying it by saying "it's all personal experience and I identify with her."

## Distrust government and the medical establishment

Some older adults used the common persuasive strategies related to distrusting government and medical establishment as heuristic cues to spot misinformation, while others fell for them. Reacting to the story about cashew is equivalent to Prozac and it is a secret that "big pharma" has been trying to hide for monetary reasons, P5 admitted that the article made the most impression on him and convinced him because "the lie parts stuck out to me. Big pharma, and big lie, that hit home." The same phrase, on the other hand, triggered skepticism for P1 during think-aloud in the middle of reading this article, "'Yet another secret that big pharma will never admit', now, my skepticism has just gone up probably threefold. . .. . .When I got to the part about that so this is a big secret–secret that big pharma is trying to hide, that's a trope I've heard before. And quite often proves to be just a way to reach people's emotions."

## Politicizing health issues

Politicizing health issues exploits individuals' existing moral values, beliefs, and identity by feeding misinformation that aligns with these values to trick them. For instance, regarding the tweet that mandatory masking is a violation of individuals' freedom and rights, P21 identified with this view: "I am concerned about freedom. . .. . .so I guess that kind of resonates with me about personal freedoms." P27 refuted: "It's using your love of church and freedom to try and promote being reckless with other people's health, in my opinion." For the radioactive pet food imported from Asia, P26 pointed out: "I think there's so many biases and stuff. I mean people who might have some bias against the different Asian countries might be like, 'Oh, that makes sense.' You know, I could see that mindset kind of playing a little bit." Some participants were particularly cognizant of the use of politics in health: "As soon as I read democratic governor, I thought, oh God, here we go. It was going to be political and they were politicizing it. So I mean the whole thing just rubbed me the wrong way." [P10]

## Highlighting uncertainty and risk

Uncertainty and risk exploit people's fear of the unknown to create an emotional state of anxiety. Some prefer safety over doubt, choosing to trust misinformation rather than dismiss it.

Regarding the radioactive pet food, P15 stated: "if they are sending this type of food, providing food makers are using this kind of food, how dangerous it is for pets as well as for humans. That's what I'm afraid of and then it talks about contaminating the entire Pacific Ocean. I think that's really dangerous, you know." P26, on the other hand, considered this highlighted risk as 'alarmist' and correctly identified this as misinformation.

### Emotional appeal

Misinformation can evoke emotions like anxiety, fear, anger, and sadness. While some older adults' were emotionally affected by these persuasive strategies, others rationally assessed the emotional appeals. For instance, regarding the misinformation that sunscreen causes breast cancer, P21 expressed sympathy: "I feel bad for that person because they, you know, it's an unfortunate situation for them." On the other hand, P18 was alert to the emotional appeal strategy: "It seems like an emotional play and somebody that needs an answer to why something happened to them specifically." Similarly, P1, quickly classified the post about radiation contaminated pet food as misinformation: "the food you purchased for your 'fur friends' can kill them, that seems like a very emotional appeal so that is a red flag."

### Exploit science's limitation

Misinformation often exploits science's inherent limitation to portray science as unstable, lacking longitudinal data, limited sample size, etc. For instance, for the misinformation on paraben in sunscreen causing breast cancer, the misinformation clicked with P12: "We all know scientific data is delayed and who knows when they may release new data, one day, retracting the previous finding. Where you hear something and then maybe six months down the road, they're saying, 'that's not necessarily true anymore'".

### Inappropriate use of scientific evidence

Health misinformation typically incorporates scientific evidence, but it is applied inappropriately to support false claims. Many were tricked by the sheer presence of scientific evidence that established legitimacy and only very few were able to see through it. For example, P24 identified the inappropriate use of scientific evidence while reading the misinformation that eating cashew is equivalent to taking Prozac: "Even though cashews probably have that compound in them, I don't think there would be enough in it to affect your depression."

### Rhetorical tricks

Rhetorical tricks in health misinformation include exaggeration and selective omission. Some participants were easily tricked without considering the omitted facts. For instance, P6 agreed with the misinformation that COVID-19 is just like a flu, stating, "it was basically saying that you know it's a 99.99% curable situation, just like the flu", without realizing that people over the age of 50 are the most vulnerable yet not included in the data. For some participants, the rhetorical trick of exaggeration put them off. Reacting to the exaggeration about the contamination of radioactive substances, P3 stated: "It seems much more of an alarmist piece than something that's really done with research."

### Biased reasoning

Misinformation may include arguments that sound reasonable but contain logical fallacies. Some fell directly for the seemingly reasonable arguments, while others were able to spot the biased reasoning. For instance, P25 stated: "Chart showing the height of the tsunami that hit

this location, that doesn't give any way to what they're trying to say about the pet food." Similarly, reacting to the post claiming the sun cream could cause breast cancer, P2 stated: "she's making a very wide sweeping argument that that particular ingredient caused her breast cancer and you can't really say that because it could be anything environmental, or possibly genetic even though she doesn't know that anybody in her family has had that type of cancer or a cancer similar to it that could turn into breast cancer."

### Distinctive linguistic features

Linguistic features, including excessive uses of uppercase words, use of extreme words and other linguistic intensifiers, can grab attention or instigate involvement, especially emotional involvement from the audience. They effectively tricked some participants. For example, P21 stated: "Well, for me it [uppercase word] just caught my attention that I just feel bad for that person because they're just making a strong statement." Yet, P28 constructively used these features to pick out misinformation: "Okay, so I'm seeing like in bold letters 'lying,' and that "this industry has been lying." So that makes me suspicious that this is not true."

### Establish legitimacy

Misinformation can deceive people by using persuasive strategies to establish legitimacy, such as citing a source even if it's fake or lacks relevant credentials, using medical jargon or words associated with health to appear credible. P16 believed the misinformation that sunscreen causes breast cancer because "she named the chemicals that were in the sunscreen." However, others (i.e., P10) questioned these legitimacy cues and indicated that further research was needed: "These are words that a regular person won't understand. If I were reading something like this, I would be looking up those words and seeing what they mean." Reacting to the post that cashews can treat depression, P27 pointed exactly to the persuasive strategy: "Natural, that's another key word that, you know, might get people in, hook them in."

### Why and how older adults fall for misinformation (RQ2)

The previous section demonstrated that responses to the same persuasive strategies in misinformation varied. Some participants were tricked by them, and some were able to spot these strategies and utilize them to conclude that misinformation was at play. So why were some participants tricked? How were they tricked? Our thematic analysis organizes factors contributing to the why and how into three categories: information factors, individual factors, and the interplay of these two factors.

### Information factors

Three characteristics of misinformation make it hard for people to discern. Firstly, there is an abundance of information. Easy access, lack of regulation, and receiving through the sharing of acquaintances, family, and friends, further complicate the cognitive challenge of older adults in distinguishing misinformation on social media. P14 said: "There's so much information you have to kind of triage it, and what's worth your time to invest in investigating it, and then need to be? Also really really careful, cuz I've gotten burned and I know lots of people where I've passed on information that turns out not to be true." With such information overload, people usually relied on heuristic cues to make it easier for them to process information, which gives leeway for persuasive strategies such as establishing legitimacy to exploit.

Second, misinformation thrives with uncertainty, especially in areas with conflicting evidence and many unknowns. P8 discussed this problem: "I have a sister who's a nurse and a

niece who is in health care as well and they're two different opposites. You want to trust your healthcare professionals. But, you know, again this is my hard time sorting it out. My sister will believe one thing, and her daughter, who was my niece, will say 'oh no, you should do it this way and you should do it this way.'"

Lastly, misinformation contains some truth but not the complete truth. Over time, the partial truth becomes dominant and is perceived as the whole truth due to easy access and sharing of information. P10 offered her explanation why the radioactive pet food misinformation may be believable, "I have heard, and I think the news has said that there has been, you know, some stuff from Japan that has floated made it over to the United States now from that tsunami. So I mean, parts of it are just believable enough that I would, that people would believe it." Similarly, regarding the post of sunscreen causing breast cancer, P9 explained: "When she talks about the paraben issue, it's a lengthy paragraph and she cites that it's a known cause of cancer and we're getting it through our largest organ. So, some of it seems to be planted with some truth or facts as far as that person believes to be."

## Individual factors

Individual susceptibility to misinformation is influenced by specific traits or the absence thereof. We identified three categories of such individual factors. First, the primary factor was the lack of media literacy, wherein individuals lack the knowledge and skills to access, critically analyze, and evaluate information, especially in diverse online platforms. P2 highlighted this vulnerability: "Sometimes it can be very difficult and sometimes you have to really dig deeper, and you don't have the time or the resources or, you know, the energies or follow it through, then you may accept it for what it is at face value, a little bit sooner than normal." Additionally, individuals who overestimate their media literacy skills are also prone to misinformation. A few participants believed that they had never encountered misinformation (e.g., P7, P16, P21) or had confidence in themselves for having critical thinking and analytical skills (e.g., P6 ["I am more like an analytical type person, I try to get all the facts, and then be logical on the facts"]), yet struggled to discern misinformation during the interview.

A second individual factor was confirmation bias, which affects everyone to some extent but poses a higher risk for those with a strong predisposition to confirmation bias. This can lead to echo chambers where "individuals seek out sources that reinforce their existing beliefs", as stated by P13. This is particularly true for identity-based motivating reasoning. If one already believes that pharmaceutical companies are evil, the government is corrupted, or natural ingredients-based medicine is better, they will be kept in the bubble and "choose to believe what they want to believe". For instance, P7 agreed that cashew was equivalent to Prozac because "I'm always looking for natural things. Like for instance, I have osteoporosis and I'm not taking anything for that either. But I do what I can out there naturally, so I think this article is right up with that."

The third trait contributing to vulnerability to misinformation was risk aversion and short-term orientation. For instance, participants were afraid of vaccines and medicines due to side effects, and they would "better be safe than sorry". The risks and side effects are usually imminent, and they overlook the long-term benefits. P16 discussed her reluctance to take the booster shot: "It sounds good, the protection, everything sounds good, but I just don't want to go through that side effects and go to the sickness."

## Interplay of individual and information factors

The interplay of information and individual factors makes people more susceptible to misinformation. When misinformation aligns with individuals' emotional needs, it can become

highly convincing. P18 commented that "very vulnerable people that are looking for answers, sometimes you have to rationalize and blame somebody for something that just happens without a reason that we know. It seems like an emotional play and somebody that needs an answer to why something happened to them specifically."

Secondly, individuals may be drawn to misinformation that justifies their existing behaviors. Regarding the misinformation that cashew is equivalent to Prozac, P8 offered her explanation: "If you like cashews, you're going to believe it because you know, this is good for me so yeah that must be."

Thirdly, misinformation can cater to individuals' wishful thinking that there is a simple cure or solution to complex problems. Using P11's words to illustrate that: "A lot of seniors are not very good. You know, as we get older, you know, oh my back's hurting and also you see something......we just, 'Oh my God that's gonna help me.'" P5 would accept that cashew could be equivalent to Prozac because: "it's always worth a try. You know, again, it is not endangering your health in any way......whether it works or not, who knows but nothing gained, nothing ventured, nothing lost or something like that". P3 commented: "who wouldn't want to eat a handful of cashews, to solve a problem."

## Manage misinformation (RQ3)

The previous sections illustrated how and why some older adults fell for the misinformation. Nevertheless, some older adults were able to identify these persuasive strategies and also discussed how they successfully managed (mis)information online. Our analysis identified the following three sub-themes.

### Manage misinformation for self: Further research when in doubt

The primary strategy mentioned by the older adults to manage (mis)information was to conduct further research. Many older adults initially had suspicions about the misinformation in this study but maintained a "truth default" mentality, leaving room for uncertainty. So many of them could not make a definite conclusion that they disagreed with the article because it was misinformation. When unsure of when the information they received contradicted their existing knowledge, they looked for additional information: "I doubt it but let's see if you can prove it to me." (P1). They verify and authenticate information through the following eight methods: 1) checking the source (e.g., "use this person, that researcher's name, and so you can do your own dig" [P26]), 2) checking cited references (e.g., "But it's nice that they link to an article from Huffington Post. You have to see what links that has within that to see if they actually have the research that they are talking about." [P17]), 3) using fact-checking websites (e.g., "if it's really something—like another one that she told me was that Moderna, was owned by Bill Gates, so like I just went to Snopes.com." [P28]), 4) looking up unfamiliar terms (e.g., "So I mean that my first thought is, I would be looking up those words and seeing what they mean." [P10]), 5) googling key words or other searchable content mentioned in the article (e.g., "I'm not aware that paraben is a carcinogen. I would want to validate that." [P1]), 6) going to legitimate and trustworthy sources for further research (e.g., "So I may look into articles on like PubMed, in terms of if cashews have been looked at for treating depression or alleviating symptoms of depression" [P2]), 7) consulting trusted individuals (e.g., "I would ask someone that's smarter than me, that's specific to this, that has more information." [P4]), and 8) using multiple online sources to verify (e.g., "Most people would say, 'I'll go to Google'. Well, Google is one. I'm going to Bing. I'm gonna go to duckduckgo."[P4]). However, participants indicated that doing further research is not an easy task. As P2 described: "I think sometimes it can be very difficult and sometimes you have to really dig deeper, and you don't have the time or the resources."

## Manage misinformation for self: Critical thinking

Participants emphasized the importance of critical thinking in managing misinformation. They highlighted the use of questioning techniques to determine the credibility of information, such as asking themselves questions such as whether they got emotional and why, what are the motives of the author, whether the source is credible, what is the meaning of specific words, especially technical words and the scope and applicability of the information. For instance, while reading the article about radioactive pet food imported from East Asia, P25 engaged in think-aloud and asked a series of questions: "I would point to things like where they are getting their information about importing seafood from waters near Fukushima. And then I would question the data from, okay, 'my friend has a Geiger counter that was inherited from her dad', so this device is years old, likely and is it still working correctly? Is it still calibrated correctly? Are they using it correctly? 'Putting it within one inch of the wet cat food.' Is a reading something else other than food? And you know what limit is set in a Geiger counter to go off?"

Secondly, deliberative counter arguments were also found to be effective in identifying fallacies in misinformation. For example, while reading the post about masking and COVID-19, P17 counterargued: "And I don't see how wearing a mask is taking away your freedom and controlling you. It seems like most churches require people to wear other type of clothing and nobody's complaining if they have to wear pants in church that their freedom and control has been taken away." However, there was also a caveat that individuals who already believed certain types of misinformation might use their existing misbelief to counter-argue true information. For instance, P5 somewhat agreed with the tweet that masking was useless and offered counter arguments for masking being useful: "I've always said if the mask is so good and why does the virus keep spreading? If the mask is that good, the virus wouldn't spread like this."

The other two types of critical thinking were: 1) acknowledging individual differences (e.g., "I guess everybody is different, cause they get, they are affected by it differently [P19]), and 2) not jumping into absolute conclusions (e.g., "I'm in between disagree and agree with the article that they are still working on it. And they just haven't come to a complete conclusion, still working on it [P16]).

## Manage misinformation for self: Actions to take upon misinformation

When faced with misinformation, participants discussed various actions to take. These included: 1) ignoring the misinformation, (e.g., "So that's why sometimes like I just see something and I just don't give it the time of day."[P14]), 2) taking steps to avoid future encounters with similar misinformation (e.g., "you could block these people, stop them from entering your feed or your, you know, dropping from your Instagram or whatever. . . . . .you know, I don't need that sort of stupidity in my life."[P22]), and 3) reporting the misinformation (e.g., "I would judge whether I thought it was presenting anything dangerous or causing people to react in a way that might inflame them to take action against other people. If I saw anything that I felt was leaning toward that, I would report the post." [P1].

## Manage misinformation for others

Besides managing misinformation for themselves, participants also discussed the need to manage misinformation for their family and friends. It was not an easy task, and not all family and friends welcomed their involvement. P10 stated: "I can try to counter them and tell them where I'm getting my information, but then they hang up on me. So, you know, so I can only do so much." They offered the following insights on when and how to effectively assist others.

First, participants suggested considering relationship closeness and deciding whether and how to engage with them, especially on social media. P10 comments: "If they were simply a

Facebook friend, I would probably just keep scrolling. If they were someone I knew well, I would probably either post it right in there, if I could find a believable source that said otherwise, I would post it right in their feed and just say 'please read this article'. If it was someone in between I would probably message them, not in the feed. I have a cousin who's a Trumper, and the posts, the things in his feed, the responses to the things in his feed are just so argumentative and so against what I believe, so I wouldn't comment on that one at all."

Second, participants attempted to demonstrate an understanding of where the person sharing the misinformation came from. Telling people they were wrong or engaging in argument was never helpful. P3 shared: "I don't want to create conflict by, you know, telling them that they're wrong, but trying to just gently guide them to look at information in a different way. And also to look from individuals' point of view, you know, because we all have our own perceptions, and we all have different experiences in our lives."

Third, participants mobilized their own critical thinking skills to gently guide their family and friends to look at information differently. Four strategies were identified. First, they shared accurate information, (e.g., "After I read to him some of the reviews and showed him a little bit more about what it was all about, he said 'no I guess you're right'". [P27]). Second, they encouraged scrutiny of misinformation (e.g., "My brother sent me something. Again, off his social media and the numbers he sent me or the quantity he sent me was so extraordinary that I wrote back to him and I said, 'Really, Think again. You really think this many, and it was a matter that the numbers were so extraordinary or the item that they stated was really kind of off the mainstream'". [P13]). Third, they encouraged others to ask questions (e.g., "I don't want to stand blames or anything like that. I want to have them at least, just question. You know I would say, 'where did you get that information along that line' or confront". [P22]). Lastly, they offered alternative explanations (e.g., "my son and his wife were double vaccinated, and they still got COVID-19, and my daughter-in-law got it very bad. So, [Son's name] is saying 'now well maybe the vaccines are useless'. And I said, 'well, you don't know because you might have died if [you didn't have a vaccine], you know.' So I try to try to explain to them" [P10].

## Discussion

The current study conducted semi-structured interviews and provided qualitative evidence from the perspective of older adults using their own words to describe how they process persuasive strategies in misinformation. Our thematic analyses revealed themes explaining why and how older adults fall for misinformation that employs persuasive strategies. The discussion will begin with informational and individual factors as well as their interplay to explain older adults' susceptibility to misinformation. Next, connections will be elaborated between the findings and media literacy suggestions that may mitigate the factors contributing to believing misinformation among older adults. Our thematic analyses also revealed major strategies that older adults engage in to manage misinformation for themselves and others. The discussion will continue with the best practices learned from them regarding misinformation management in everyday life.

### Why and how older adults fall for misinformation with persuasive strategies

Among the twelve groups of persuasive strategies in misinformation, our findings showed that some older adults detected the persuasive strategies as red flags, while others were tricked by them. Persuasive strategies that grab attention, such as narrative, personal stories, emotional appeals, and distinctive linguistic features, not only make the misinformation stand out in a sea of information but also draw the audience in due to cognitive ease and emotional

engagement [22]. These strategies tend to work on those who do not critically analyze information or engage in systematic processing [48]. Additionally, in areas where there is abundance of unknown and conflicting evidence, misinformation is easy to get around. For instance, persuasive strategies such as exploiting science' limitations, and highlighting risk and uncertainty may increase participants' doubts about science and be drawn into conspiracy theories [14, 24]. The last information factor is the partial truth in misinformation. Among the twelve groups of persuasive strategies, inappropriate use of scientific evidence, rhetorical tricks, and biased reasoning all contain some truth to mislead the audience. For instance, general scientific principles may be referenced, but no evidence was available for the specific domain [16] or for providing factual information and connecting the pieces of information without valid reasoning [24]. These strategies in misinformation are tough to tackle because while the information is factual, the reasoning behind is flawed. Individuals with certain traits such as low cognitive abilities or a tendency to believe in truth by default are more susceptible to these persuasive strategies [49, 50].

## Improving media literacy to mitigate susceptibility to persuasive strategies

Our findings have shown evidence of older adults falling for the common persuasive strategies while processing online health misinformation. To mitigate their susceptibility to the common persuasive strategies, older adults should be equipped with the knowledge of what they are and be able to identify the common tricks and pitfalls. Media literacy encompasses the knowledge and skills that people need to become mindful media content consumers [51, 52]. The knowledge and skills to identify common persuasive strategies in online health misinformation should be a critical part of media literacy education for older adults. Research has shown that older adults are more susceptible to online misinformation not due to their declining cognitive abilities but rather because they have less experience or knowledge with online information [10], which may lead older adults to apply truth default while consuming online information [45].

Prior research has attempted to include tips for spotting misinformation in media literacy programs, primarily focusing on formatting cues (e.g., misspellings, awkward layouts) and direct behavioral suggestions (e.g., inspecting the dates and sources) [52]. However, given the increasing prevalence of high-quality misinformation, it's more crucial to understand how persuasive arguments are created. For example, evidence-based misinformation is seen as more accurate than fact-free misinformation [53]. Recent AI-generated misinformation makes use of sophisticated persuasive strategies such as enhancing narrative details, simulating emotional appeals, and making conclusions using biased reasoning [54]. Simple behavioral suggestions like checking evidence or sources are insufficient in navigating today's complex media environment [55]. Learning common persuasive strategies underpinning online health misinformation is pivotal in helping older adults critically analyze content and arguments.

One caveat is that the knowledge and skills to identify persuasive strategies in online misinformation are not necessarily easy to learn and may need conditions to practice in everyday information consumption. However, once the knowledge and skills are learned, the benefits can be sustainable in the long term. Learning and practicing persuasive strategies require the motivation of older adults to spend time and cognitive resources. According to the dual-process theories such as the Elaboration Likelihood Model (ELM), people are likely to scrutinize message arguments when they find the issue is personally important and relevant [56]. Similarly, literature has revealed that people who have analytic thinking styles show more acceptance of corrective messages explaining why a piece of misinformation is incorrect [57]. The media literacy program teaching persuasive strategies may be more acceptable and successful among older adults who recognize the importance of discerning online misinformation and

who have greater needs for cognition. However, it is worth pointing out that some of the persuasive strategies could be identified as cues that only require peripheral information processing. For example, people can apply limited cognitive processing to spot personal anecdotes and emotional appeals such as emotionally intensive words [58]. The media literacy program may begin with teaching simple persuasive strategies that only require peripheral information processing and deepen the education with more complex persuasive strategies (e.g., biased reasoning). Additionally, as prior research found that media literacy training for older adults is better learned when taught in a social context where family and friends are involved [59], media literacy programs to help older adults to manage misinformation can also benefit from social facilitation.

In addition, it is important to create awareness of how media content interacts with individual vulnerabilities, particularly in relation to misinformation. Emotional and wishful thinking are identified as common vulnerabilities that can make people more susceptible to misinformation. When misinformation triggers people's emotions such as anger and anxiety, people's information processing will be masked by such emotions. A recent review has concluded that anger and greater cortisol response to stress increased people's vulnerability to misinformation [60]. Media literacy education should focus on critical engagement with media contents by encouraging people to reflect on their emotions and detach it from credibility assessment while processing information. It will be helpful for people to recognize their emotions at the moment of consuming media content, reflect on how media content may fill in their emotional needs, and assess information credibility based on its accuracy rather than its ability to make them feel better [61]. With respect to health misinformation, this media literacy education may be particularly important for health issues, such as vaccines, that tend to be politicized and may trigger affect polarization [62]. An understanding of how strong emotion may impair our ability to rationally process science information provides a foundation for people to manage their emotions when they are motivated to discern misinformation.

### Managing misinformation for self

Similar to prior research [39], during the initial encounter with an information item, our older adult participants engaged in internal authentication acts involving the assessment of information credibility based on the source, the message, and their prior experience and knowledge. Researching information further was a common strategy to manage online health misinformation, driven by suspicions about the legitimacy of the source and original research of the evidence cited in the information. However, not everyone pursues further research, influenced by belief in misinformation's accuracy or a perceived lack of importance. For instance, prior research found that older adults sometimes resorted to the strategy of avoiding social media content [63, 64]. Although prior research found that people verify information when doubting its accuracy or when confirming existing beliefs [39], recent research shows that the predictor of people's motivation to verify information is how much they care about the information or how important the information is to them [65].

For people who are willing to conduct verification, further information search is not always easy and does not guarantee an accurate assessment of misinformation. Online health information requires a high reading level to understand complex health issues that involve conflicting evidence [66]. For some older adults, they may turn to trusted family members to assist them [67]. Our older adults participants shared the following strategies to develop the competence of navigating through the complex information environment. The first aspect is related to how to conduct further research. Our findings suggested effective practices, including checking the source and references cited in the information, using search engines to verify

details, using fact-checking websites to verify facts, looking up unfamiliar terms, going to legitimate and trustworthy sources for further research, and verifying with trusted individuals. These practices for future research are effective but may provide other layers of complexity and challenges. For example, using search engines to verify details requires the capability of selecting, analyzing, and integrating information from various sources [53]. Additionally, choices of trustworthy sources of information may be subjective to individuals' political ideology [68].

Another competence to develop is critical thinking, such as practicing generating alternative and/or counterarguments to claims. When examining evidence, reasoning, and conclusions, people can be encouraged to create counterarguments at each step. Does the evidence support an alternative reasoning and does the reasoning lead to an alternative or an opposing conclusion? If yes, what evidence or other information is needed to make a better decision? These insights may be incorporated in future media literacy interventions to educate older adults how to react to media content and in future information literacy interventions to strengthen older adults' ability in finding, evaluating, and using information [69]. Moreover, when examining personal experience and stories, individuals should consider individual differences and the extent to which personal experience can be generalized. The experience described by a person may be true to the person, but what backgrounds or other contextual factors may have shaped the experience. They can ask themselves questions: Does the person or the storyteller draw a correct conclusion from his/her experience? Can the person's case be generalized to others? These are the questions to ask while processing information.

## Managing misinformation for others

The closeness of the relationship plays an important role as a moderator in terms of how people manage misinformation for others. Our findings are consistent with the existing literature that relationship closeness [38] and the availability of channel affordances [37] impact individuals' decision about whether or not to correct or challenge misinformation from others. A pattern found in the current study is that they would first think about how close they were to a target person and then choose a desired communication channel (e.g., private text message vs. confrontation in face) to talk about sensitive issues. Participants also mentioned that they would respond to misinformation from strangers online by posting a counterarguing comment. However, when misinformation comes from friends and or family members, they would choose a less face-threatening way (e.g., private text) that does not damage interpersonal relationships [38]. Moreover, when misinformation comes from a close tie, people may avoid discussing misinformation at all or gently suggest alternative information sources rather than confronting the person with counterarguments [43].

In terms of how to discuss misinformation with others, our findings reveal that showing understanding and empathy before challenging or correcting misinformation is important, as it facilitates persuasion and reduces psychological reactance, which is likely to occur when discussing misinformation [70]. The feeling of being understood and recognized could reduce perceived threat from someone's challenges or corrections of misinformation. Therefore, we recommend practicing empathetic communication when managing misinformation for others. Future research may also examine this correction approach beyond the interpersonal communication setting to other health settings [55].

Lastly, our findings make another contribution to the literature by suggesting nudging strategies used to manage misinformation for family and friends. Four nudging strategies were identified: providing true information and sources, prompting scrutiny of misinformation, encouraging questions, and providing alternative explanations. The four strategies may be

used depending on others' individual characteristics, such as personality and their relationship closeness [40].

## Limitation

The study's findings should be interpreted considering the following limitations. First, we included older adults aged at least 50 (participants' ages ranged from 51 to 76 with an average age of 62) and most of whom had higher education and used Zoom to participate in this study. This group of older adults were relatively more technologically savvy and may have high media and digital literacy compared to the average older adults. Given that this study focused on online health misinformation, the group of participants represented the type of older adults more likely to use the Internet for health information and thus the implications of online health misinformation were more relevant to them. However, the findings may not be transferable to other older adults not represented by our sample. Second, the participants may be more alert and cognizant when processing the information encountered in this study than they usually would in their everyday life. Nevertheless, our findings demonstrated that these persuasive strategies were still quite effective, suggesting that it is important to educate the public to see through these strategies in misinformation. Third, although we used a systematic approach to creating the misinformation messages, i.e., embedding persuasive strategies identified in a systematic review to misinformation contents that were found from fact-checking websites and using various popular online platforms to present the information to the participants, the messages may not be representative of all types of online health misinformation and future research is needed to expand the current study.

## Conclusion

This study provides qualitative evidence on why and how older adults fall for misinformation crafted with persuasive strategies at the moment of encountering them. Thematic analysis demonstrated factors related to the information content, the individual, and their interplay, which may contribute to misinformation susceptibility. Misinformation management tactics were extracted from those individuals who could see through the persuasive strategies embedded in the misinformation content. The insights of the findings can inform the design of future digital literacy programs to help individuals, especially older adults, combat misinformation.

## Supporting information

**S1 Appendix. Four health misinformation articles with annotated persuasive strategies.**
(DOCX)

**S2 Appendix. Six messages participants exposed to.**
(PDF)

**S3 Appendix. Interview guide.**
(DOCX)

## Acknowledgments

The authors would like to thank Sue Lim and Huiyi Liu for assisting in conducting interviews and coding the transcripts.

## Author Contributions

**Conceptualization:** Wei Peng, Jingbo Meng.

**Formal analysis:** Wei Peng, Jingbo Meng.

**Investigation:** Wei Peng, Jingbo Meng.

**Methodology:** Wei Peng.

**Writing – original draft:** Wei Peng, Jingbo Meng, Barikisu Issaka.

**Writing – review & editing:** Wei Peng, Jingbo Meng, Barikisu Issaka.

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
