## [Decision Letter · Decision Letter 0]

16 Jan 2024

PONE-D-23-40641Navigating persuasive strategies in online health misinformation: An interview study with older adultsPLOS ONE

Dear Dr. Peng,

Thank you for submitting your manuscript to PLOS ONE. After careful consideration, we feel that it has merit but does not fully meet PLOS ONE’s publication criteria as it currently stands. Therefore, we invite you to submit a revised version of the manuscript that addresses the points raised during the review process.

**ACADEMIC EDITOR: **

Dear Authors,

Thank you for submitting your manuscript to PLOS ONE. After careful consideration of two independent reviews, it is clear that while your study on older adults' navigation of persuasive online health misinformation is both timely and significant, major revisions are necessary before it can be considered for publication.

The reviewers have identified several key areas that require substantial improvement. These include but are not limited to, the clarity of your research questions, methodological details, and the depth of your discussion section. Your findings offer valuable insights; however, the current structure and presentation of your manuscript do not fully convey the potential impact of your work.

To progress towards publication, it is essential that you address all the points raised by the reviewers in a comprehensive and detailed manner. This revision is not just a matter of making minor adjustments, but rather it involves a significant restructuring of your manuscript to meet the journal's standards.

We recognize the effort and dedication you have put into your research and believe that with thorough revisions, your study could make a meaningful contribution to the field. We look forward to receiving your revised manuscript.

We look forward to receiving your revised manuscript.

Kind regards,

Nicola Diviani

Academic Editor

PLOS ONE

2. In the ethics statement in the Methods, you have specified that verbal consent was obtained. Please provide additional details regarding how this consent was documented and witnessed, and state whether this was approved by the IRB.

 [Brandt Fellowship].  

Reviewers' comments:

Reviewer's Responses to Questions

**Comments to the Author**

1. Is the manuscript technically sound, and do the data support the conclusions?

Reviewer #1: Partly

Reviewer #2: Partly

2. Has the statistical analysis been performed appropriately and rigorously? 

Reviewer #1: N/A

Reviewer #2: N/A

3. Have the authors made all data underlying the findings in their manuscript fully available?

Reviewer #1: No

Reviewer #2: No

4. Is the manuscript presented in an intelligible fashion and written in standard English?

Reviewer #1: No

Reviewer #2: Yes

5. Review Comments to the Author

Reviewer #1: Thank you for the opportunity to read the article entitled ‘Navigating persuasive strategies in online health information: An interview study with older adults’. The manuscript describes a qualitative interview study about how older adults identify and cope with online health-related misinformation. The manuscript addresses an important topic from a qualitative angle, and provides rich data and potentially in-depth insights in how older people attend to and cope with online misinformation. It is clear from the manuscript that the authors have put a lot of effort in the data collection, data analysis and report of the study. I find the topic interesting and important and believe that the richness of the qualitative data can add something to the literature on online health-related misinformation attention, awareness and effects. At the same time I have some questions and concerns which I will mention point by point below, and I hope that they are helpful to the authors in the strengthening of their manuscript.

Major points

- There is a lot of information in this manuscript – which is common for a qualitative study. Therefore, a clear structure is vital to guide reader through the manuscript and understand the focus of the study. The most easy way to do this - in my view is - to formulate clear aims and research questions and use those to clearly structure the manuscript. The current manuscript formulates research questions and aims, but their relation with how the rest of the manuscript is structured (e.g., the theory section and results section) was not always clear to me. As a result I felt often lost while reading as so many topics are discussed.

- The title mentions ‘persuasive strategies’ which is a central theme to the study. These strategies have been identified in a prior literature review conducted by the authors. I find this classification interesting, and unfortunately due to anonymity I cannot check the article they refer to. The twelve strategies are central to the first part of the literature review. In my view, the 12 strategies would better fit the method section (e.g., in the form of a text box) as they are clearly part of the operationalization of the study. I assume that the strategies have been used to develop the messages that the participants were presented with, or that at least that the strategies can be identified in the messages. The literature review, I would expect to discuss more board themes/categories of misinformation characteristics (in contrast to the individual strategies that were identified in a single study). For example in terms of layout/format, content, role of scientific evidence, to place the strategies in a broader context of the literature.

- The literature review could perhaps focus more on older adults in particular and what is known about this group in terms of encountering, attending, and responding to (e.g., reasons to check or share) health-related misinformation online and a discussion of the distinction between misinformation (can be false information, but unintendedly or being outdated) and disinformation (intendedly false information).

- I was surprised that some theoretical mechanisms that are important to misinformation effects, such as confirmation bias, are only mentioned in the results section and not in the literature review section (as a reason why people believe misinformation).

- At some points, some rather strong conclusions are drawn upon this qualitative study and I think that the authors could be a bit more careful in their formulation. For example the conclusion on page 21 mentions that “based on the findings media literacy is a solution to mitigate misinformation susceptibility”. As media literacy is not measured or assessed in some way in this study, this should perhaps be formulated more carefully. Another example is the statement on page 208 (misinformation usually incorporates the trope of big pharma being evil). A reference would be helpful here. I also wonder whether this statement refers to misinformation or mainly disinformation.

- Some critical information about the study is missing in the methods section. These are for example, information about how and where the participants of this study were recruited, when participants qualified as ‘older adults’ – what was the minimum age? Did other inclusion/exclusion criteria apply? In which country was the study conducted? I assume it was in the US but would be good to mention. Who conducted the interviews?

- It would be informative to see the articles that were discussed with the participants added as appendix to the manuscript. This would allow the reader to see how the various strategies were incorporated in each article. Am I correct that all 12 persuasive strategies were incorporated in each article? How was this done? What was the length of the articles? What role played images (if they were there)? Where they all designed in the form of a Facebook post? (line 203).

- I also wonder whether the authors tested the knowledge of the participants on the topics of the messages prior to the study. I can imagine that peoples prior knowledge might also (or mainly) impact people’s ability to sport misinformation, perhaps more than the persuasive strategies within the articles.

- There is no limitation section in the article.

Minor points

- Throughout the manuscript, it says “12 groups of persuasive strategies”. This is a bit confusing, because 12 is already a lot and the word groups implies that each strategy consists of multiple strategies. In the discussion section on page on page 12 (line 475) the word group is left out, which I find more logical and clear.

- It is mentioned that the interviews were part of a larger study and that other parts of the findings are reported elsewhere. For reasons of transparency I would advise the authors to provide references to related studies here.

- The example questions from the interview guide mentioned on page 7/8 are helpful. The authors might consider adding the entire interview guide as an appendix.

- Who did the coding of the interviews? Did the first and second coder code the interviews independently?

Reviewer #2: Thanks for the opportunity to review this manuscript which delves into the perspective of older adults while navigating persuasive online health misinformation. The attention on the specific population subgroup is needed, and the methodological approach is adequate. I believe the manuscript could be improved by addressing the comments listed below.

Title: I would encourage the authors to try finding a more informative title.

Abstract Revision:

It would be beneficial to consider explicitly stating the research gap that the study addresses at the beginning of the abstract, as this would provide readers with a clearer understanding of the study's significance.

Methods Section:

Provide a more detailed explanation of the analysis process, including the coding procedures and criteria.

Clarify the reasons behind the reporting decision by stating that it was made to focus on the most relevant results while ensuring the paper's conciseness.

Include a table summarizing participant characteristics, and expand on population, sampling, and recruitment procedures.

Empirical Study Design:

Offer more information about the design and selection of the misinformation used in the empirical study. Explain how this misinformation was chosen and its relevance to the research questions.

Results Section:

Consider referencing more specific raw data or examples from the study to support the results and enhance their clarity.

Discussion Section:

Add an introductory section to the discussion to provide context and set the stage for the subsequent points.

Elaborate on how the findings relate to the idea of "improving media literacy," discussing its sustainability and feasibility in more depth.

Explore the significance of individuals being able to recognize their emotions, considering its implications for health misinformation discernment.

Incorporate discussions about relevant dual-process theories like the Elaboration Likelihood Model (ELM) to better explain cognitive processing in the context of misinformation.

Delve into the complexities and challenges associated with developing the competences necessary for discerning conflicting health evidence.

Connect the influence of personal relationships as a moderator of misinformation's impact with the respective results to provide a clearer link.

Minor Edits:

Follow the WHO naming conventions by using "COVID-19" instead of "Covid-19" throughout the paper.

6. PLOS authors have the option to publish the peer review history of their article (what does this mean?). If published, this will include your full peer review and any attached files.

Reviewer #1: No

Reviewer #2: No

---

## [Author Response · Author response to Decision Letter 0]

10 Mar 2024

The response letter to reviewers is attached in the submission.

---

## [Decision Letter · Decision Letter 1]

28 May 2024

PONE-D-23-40641R1Navigating persuasive strategies in online health misinformation: An interview study with older adults on misinformation managementPLOS ONE

Dear Dr. Peng,

Thank you for submitting your manuscript to PLOS ONE. After careful consideration, we feel that it has merit but does not fully meet PLOS ONE’s publication criteria as it currently stands. Therefore, we invite you to submit a revised version of the manuscript that addresses the points raised during the review process.

Dear Authors,

Thank you for your detailed revision of the manuscript. The paper has significantly improved as a result. However, one of the original reviewers (Reviewer #2) has a few outstanding concerns that need to be addressed before we can proceed with publication. I look forward to receiving a revised version that addresses these issues.

Best regards,

Nicola Diviani

We look forward to receiving your revised manuscript.

Kind regards,

Nicola Diviani

Academic Editor

PLOS ONE

Journal Requirements:

Reviewers' comments:

Reviewer's Responses to Questions

**Comments to the Author**

1. If the authors have adequately addressed your comments raised in a previous round of review and you feel that this manuscript is now acceptable for publication, you may indicate that here to bypass the “Comments to the Author” section, enter your conflict of interest statement in the “Confidential to Editor” section, and submit your "Accept" recommendation.

Reviewer #2: All comments have been addressed

Reviewer #3: All comments have been addressed

2. Is the manuscript technically sound, and do the data support the conclusions?

Reviewer #2: Yes

Reviewer #3: Yes

3. Has the statistical analysis been performed appropriately and rigorously? 

Reviewer #2: N/A

Reviewer #3: Yes

4. Have the authors made all data underlying the findings in their manuscript fully available?

Reviewer #2: Yes

Reviewer #3: Yes

5. Is the manuscript presented in an intelligible fashion and written in standard English?

Reviewer #2: Yes

Reviewer #3: Yes

6. Review Comments to the Author

Reviewer #2: The manuscript in its present form has improved and most of the comments from reviewers have been adequately addressed.

However, I still believe that even if the introduction section is quite extensive, the reasons justifying the interest on the specific subpopulation (older adults) are not adequately presented. I would suggest dedicating a section to this and also reflecting specific considerations into the discussion.

I noticed a typo on page 2 line 60.

Reviewer #3: (No Response)

7. PLOS authors have the option to publish the peer review history of their article (what does this mean?). If published, this will include your full peer review and any attached files.

Reviewer #2: No

Reviewer #3: No

---

## [Author Response · Author response to Decision Letter 1]

6 Jul 2024

Response: We now have dedicated a section in the introduction to justify focusing on this specific subpopulation of older adults (the 3th paragraph of the Introduction). 

We also added in the discussion section reflecting specific considerations of older adults (p. 28 and p. 30).

We also carefully proofread the manuscript for grammar and typos. Thank you!

---

## [Editor Report · Decision Letter 2]

11 Jul 2024

Navigating persuasive strategies in online health misinformation: An interview study with older adults on misinformation management

PONE-D-23-40641R2

Dear Dr. Peng,

We’re pleased to inform you that your manuscript has been judged scientifically suitable for publication and will be formally accepted for publication once it meets all outstanding technical requirements.

Kind regards,

Nicola Diviani

Academic Editor

PLOS ONE
---

## [Editor Report · Acceptance letter]

15 Jul 2024

PONE-D-23-40641R2 

PLOS ONE

Dear Dr. Peng, 

I'm pleased to inform you that your manuscript has been deemed suitable for publication in PLOS ONE. Congratulations! Your manuscript is now being handed over to our production team.

Kind regards, 

on behalf of

Dr. Nicola Diviani 

Academic Editor

PLOS ONE